# Leveraging affinity cycle consistency to isolate factors of variation in learned representations

## Abstract

Identifying the dominant factors of variation across a dataset is a central goal of representation learning. Generative approaches lead to descriptions that are rich enough to recreate the data, but often only a partial description is needed to complete downstream tasks or to gain insights about the dataset. In this work, we operate in the setting where limited information is known about the data in the form of groupings, or set membership, and the task is to learn representations which isolate the factors of variation that are common across the groupings. Our key insight is the use of *affinity cycle consistency* (ACC) between the learned embeddings of images belonging to different sets. In contrast to prior work, we demonstrate that ACC can be applied with significantly fewer constraints on the factors of variation, across a remarkably broad range of settings, and without any supervision for half of the data. By curating datasets from Shapes3D, we quantify the effectiveness of ACC through mutual information between the learned representations and the known generative factors. In addition, we demonstrate the applicability of ACC to the tasks of digit style isolation and synthetic-to-real object pose transfer and compare to generative approaches utilizing the same supervision.

## 1 Introduction

Isolating desired factors of variation in a dataset requires learning representations that retain information only pertaining to those desired factors while suppressing or being invariant to remaining "nuisance" factors. This is a fundamental task in representation learning which is of great practical importance for numerous applications. For example, image retrieval based on certain specific attributes (e.g. object pose, shape, or color) requires representations that have effectively isolated those particular factors. In designing approaches for such a task, the possibilities for the structure of the learned representation are inextricably linked to the types of supervision available. As an example, complete supervision of the desired factors of variation provides maximum flexibility in obtaining fully disentangled representations, where there is a simple and interpretable mapping between elements and the factors of the variation (Bengio et al., 2013). However, such supervision is unrealistic for most tasks since many common factors of variation in image data, such as 3D pose or lighting, are difficult to annotate at scale in real-world settings. At the other extreme, unsupervised representation learning makes the fewest limiting assumptions about the data but does not allow control over the discovered factors of variation.

The challenge is in designing a learning process that best utilizes the supervision that can be realistically obtained in different real-world scenarios. In this paper, we consider weak supervision in the form of set membership (Kulkarni et al., 2015; Denton & Birodkar, 2017). Specifically, this weak set supervision assumes only that we can curate subsets of training data where only the desired factors of variation to be isolated vary, and the remaining nuisance factors are fixed to same values. We will refer to the factors that vary within a set as the *active* factors, and those that have fixed and same values as *inactive*. To illustrate this set supervision, consider the problem of isolating 3D object pose from images belonging to an object category (say, car images). The weak set supervision assumption can be satisfied by simply imaging each object from multiple viewpoints. Note, this would not require consistency or correspondence in viewpoints across object instances, nor any target pose val-

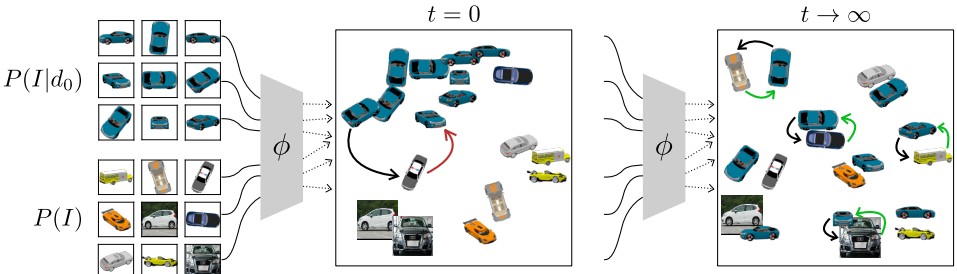

Figure 1: *Affinity cycle consistency (ACC) yields embeddings which isolate factors of variation in a dataset $P(I)$. It leverages weak supervision in the form of set membership, such as in the set of images $P(I|d_0)$ rendered around a given synthetic car (top, left). A cycle consistency loss encourages finding correspondence between sets of inputs by extracting common factors that vary within both sets and suppressing factors which do not. We show that ACC isolates nontrivial factors of variation, such as pose in the example above, even when only one of the sets has been grouped. Importantly, this allows the incorporation of data with no supervision at all, such as the images of real cars (bottom, left). The learned representations (right, $t \to \infty$), contain only the isolated factor of variation (contrast the alignment here with the untrained representations shown in the middle, $t = 0$).*

ues attached to the images. In practice, collecting multiple views of an object in a static environment is much more reasonable than collecting views of different objects with identical poses.

In this paper we propose a novel approach for isolating factors of variation by formulating the problem as one of finding alignment between two sets with some common active factors of variation. Considering the application of synthetic-to-real object pose transfer, Figure 1 illustrates two sample sets of car images where pose is the only active factor in the first set $P(I|d_0)$ of synthetic car images and the second set $P(I)$ is comprised of both real and synthetic car images. Given these sets, without any other supervision, the aim is to automatically learn the embeddings that can find meaningful correspondences between the points in the two sets. The key idea behind our approach is a novel utilization of cycle consistency. A cycle consistent mapping can be described broadly as some non-trivial mapping that brings an input back to itself, and in our case the mapping is between sets of points in embedding space. We denote our application of cycle consistency as *affinity cycle consistency* (ACC) as it uses a differentiable version of soft nearest neighbors since the correspondence forming the cycle or not known *a priori*[1]. Further, no explicit pairwise correspondence between the input sets is needed; it is found by the loss. We posit that this process of finding correspondences is crucial to isolating the desired factors of variation: to match across sets, the representations must ignore commonality within a set (the inactive factors) and focus on the active factors common to both the sets. For example, ACC-learned embeddings from the two sets of car images in Figure 1 can isolate the object pose factor as that is the common active factor across both the sets.

We also show how our ACC model can be generalized to the partial set supervision setting: ACC can learn to isolate factors of variation even when set supervision is provided for only one set, while the second set is virtually unrestricted. This has practical importance as it allows us to integrate unsupervised data during training. In Section 4.3 we show how this process can be applied to isolate 3D pose in real images without ever seeing any supervised real images during training.

In the following two sections we cover the related works and formally introduce our ACC method. Given the novelty of our approach for isolating factors of variation, we present a progression of experiments to develop an intuition for the technique as it operates in different scenarios. In Section 4.1 we evaluate ACC in various settings using the synthetic Shapes3D dataset where the latent factor values are known, allowing a quantitative analysis. Later, in Section 4.2 we demonstrate the use of ACC in isolating handwritten digit style from its content (class id). In Section 4.3, we show how ACC can be applied in its most general form to isolate 3D object pose in real images with a training

---

[1]This specific loss has been used previously in Dwibedi et al. (2019) to align different videos of the same action, and here we show this loss is much more general. We term it *affinity cycle consistency* as opposed to the prior work's terminology, *temporal cycle consistency*, to indicate as such.

process that combines a collection of set-supervised synthetic data with unsupervised real images. We conclude with a discussion and analysis.

## 2 RELATED WORK

**Disentangled representations.** Most approaches toward disentangled resprepresentations are unsupervised, and are generally based on generative modeling frameworks such as variational autoencoders (Kingma & Welling, 2014) or generative adversarial networks (Goodfellow et al., 2014). The VAE is a latent variable model that encourages disentanglement through its isotropic Gaussian prior, which is a factorized distribution. Numerous variations of the VAE have been proposed to further disentanglement, and these include $\beta$-VAE (Higgins et al., 2017), $\beta$-TCVAE (Chen et al., 2018), FactorVAE (Kim & Mnih, 2018), DIP-VAE (Kumar et al., 2018), JointVAE (Dupont, 2018), and ML-VAE (Bouchacourt et al., 2018). InfoGAN (Chen et al., 2016) encourages an interpretable latent representation by maximizing mutual information between the input and a small subset of latent variables. In Hu et al. (2018) adversarial training is combined with mixing autoencoders. In Locatello et al. (2019) it is shown that true unsupervised disentanglement is impossible in generative models, and inductive biases or implicit supervision must be exploited. Supervision has been incorporated in different ways. Graphical model structures are integrated into the encoder/decoder of a VAE to allow for partial supervision (Siddharth et al., 2017). In Sanchez et al. (2020) disentanglement without generative modeling is proposed employing similar set supervision, but requires sequential training to learn all the factors of variation so that those varying across a set may be encoded. In contrast to these approaches, ACC produces entangled representations which directly target and isolate factors of variation complementary to those for which the set supervision is known.

**Cycle consistency.** Often, cycle consistency has been used as a constraint for establishing point correspondences on images (Zhou et al., 2016; Oron et al., 2016) or 3D point clouds (Yang et al., 2020; Navaneet et al., 2020). In a different setting, the time window between the image frames of Atari games can be learned using a discrete version of cycle consistency (Aytar et al., 2018). In contrast to Aytar et al. (2018), we use a discriminative approach and do not recover a disentangled representation. Cycle consistency has also been used in disentangling factors of variation with variational autoencoders using weak supervision in the form of set supervision (Jha et al., 2018). The most closely related work to ours is the work on temporal cycle consistency (TCC) (Dwibedi et al., 2019), where a differential soft nearest-neighbor loss is used to find correspondences across time in multiple videos of the same action. Our approach differs in two key ways. First, we generalize the approach to a broader class of problems where the data does not provide an ordering and is less likely to permit a 1-1 correspondence between sets. Second, and most significantly, we show the fundamental constraint of TCC, that both sets must share common active factors of variation, can be relaxed to allow one unconstrained set (no inactive factors), which allows for incorporating training data with no set supervision.

**Weak supervision.** For recovering semantics, there exists several self-supervision methods (Chen et al., 2020; Misra & van der Maaten, 2019) that rely on elaborate data augmentation and self-supervision tasks such as jigsaw and rotations. Augmentation can be effective if it is known how factors of variation act on the image space, but this is only true for some fully observable factors such as 2D position and orientation. This restriction similarly applies to models that bake transformations into the architecture, as with spatial transformers (Hinton et al., 2011) or capsules (Jaderberg et al., 2015). Often, we use fully self-supervised methods for recovering semantics necessary for downstream tasks. In this work, we are interested in isolating geometric factors of variation such as pose that are difficult to annotate. In order to do this, we rely on set supervision (Kulkarni et al., 2015; Mathieu et al., 2016; Cohen & Welling, 2015). In Cohen & Welling (2015) the latent representations of the training images are optimized limiting view synthesis to objects seen at training time. In practice, getting full supervision with ground truth parameters for geometric entities such as lighting and pose is challenging. On the other hand, one can often capture videos where we fix one or more of these factors of variation, and allow the others to vary. This form of set supervision is a good tradeoff between labor-intensive manual annotation required for fully supervised methods, and fully self-supervised methods.

**3D pose aware representations.** An important factor of variation for many image tasks is 3D object pose, and not surprisingly there have been attempts to learn representations which encode this property. The $SO(3)$-VAE (Falorsi et al., 2018) places a uniform prior on the 3D rotation group $SO(3)$, which allows learning manifold-valued latent variables. Latent representations that are pose-

equivariant have been proposed in Worrall et al. (2017); Rhodin et al. (2018), and this allows for pose to be directly transformed in the latent space. Generative techniques for pose disentanglement include Kulkarni et al. (2015); Yang et al. (2015a). In Kulkarni et al. (2015) a simplistic experimental setting is considered (fewer factors of variation with synthetic or grayscale images, and disentangling only the 1D azimuth angle).

# 3 AFFINITY CYCLE CONSISTENCY

**Notation:** We are given a finite set of $n$ training samples $\mathcal{S} = \{x_1, \ldots, x_n\}$, where $x_i \in \mathcal{X}$, and $\mathcal{X}$ denote the instance space with dimensions $D$. Let $\phi(x, w) : \mathcal{X} \to \mathbb{R}^E$ be the function that maps the input vector to an embedding in $E$-dimensional space. We use an affinity cycle consistency loss on the embeddings $\ell : \mathbb{R}^E \to \mathbb{R}_+$ to learn the parameters: $w^* = \arg\min_w \mathbb{E}_{x \sim \mathcal{X}} \ell(\phi(x, w))$.

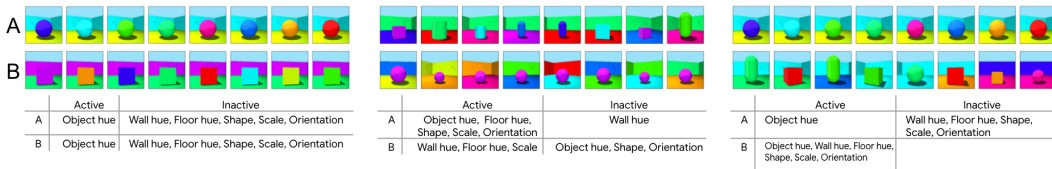

Figure 2: ***Sets with different active and inactive factors of variation from the Shapes3D dataset** (Burgess & Kim, 2018). On the left, we have the same active factor of variation for both sets. In the middle, we have some overlapping active factors of variation. On the right, we have no known inactive factor of variation for set $B$. We find* ACC *to be effective in all these three scenarios.*

**Mini-batch construction using set-membership:** Our goal is to learn $\phi : \mathcal{X} \to \mathbb{R}^E$ such that the learned representations isolate certain desired factors of variation and are invariant to certain other factors of variation. Core to the method is how the data is grouped: the affinity cycle consistency loss operates on sets, and strives to obtain 1-1 correspondence between two sets of points in the embedding space (Dwibedi et al., 2019). In the experiments of this paper, we either leverage natural groupings of images or, in the pursuit of insight, curate images into sets for mini-batch construction.

Let us refer to the two sets of images for a particular mini-batch by $A \subset \mathcal{X}$ and $B \subset \mathcal{X}$. In Figure 2, we show sets with several active and inactive factors of variation from the Shapes3D dataset (Burgess & Kim, 2018). In the most restrictive setting, both sets have identical active and inactivate factors of variation as shown in Figure 2(left); this was the case of (Dwibedi et al., 2019). ACC also functions in less restrictive scenarios where a subset of active factors of variation are shared between sets, as in Figure 2(middle), to the extreme where one of the sets is completely unconstrained without any inactive factors of variation, as in Figure 2(right).

Let the images in each of these sets be given by $A = \{a_1, \ldots, a_n\}$ and $B = \{b_1, \ldots, b_m\}$, respectively. Let us denote the associated embeddings as $L = \{l_1, \ldots, l_n\}$ and $M = \{m_1, \ldots, m_m\}$, where $l_i = \phi(a_i, w)$ and $m_i = \phi(b_i, w)$. Functionally, we parameterize $\phi$ with the same neural network for both $A$ and $B$. Let $d(x, y)$ denote a distance metric between points in embedding space. The notion of cycle consistency is used in many different contexts, and we use the following definition.

**Definition 1 (*Cycle consistency*)** *Given two sets of points* $L = \{l_1, \ldots, l_n\}$ *and* $M = \{m_1, \ldots, m_m\}$, *we say that* $l_i \in L$ *is cycle consistent if* $l_i = \arg\min_{l \in L} d(m, l)$ *where* $m = \arg\min_{m \in M} d(l_i, m)$.

In other words, we say that a point $l_i \in L$ is cycle consistent if its nearest neighbor $m \in M$ also has $l_i$ as its nearest neighbor in $L$. In order to use cycle consistency as our loss function, we will use a differentiable formulation of the cycle consistency loss. To get a differentiable loss, we approximate the nearest neighbor with soft nearest neighbor (Goldberger et al., 2004), which has also been used in many applications (Movshovitz-Attias et al., 2017; Rocco et al., 2018; Snell et al., 2017).

**Definition 2 (*Soft nearest neighbor*)** *Given two sets of points* $L = \{l_1, \ldots, l_n\}$ *and* $M = \{m_1, \ldots, m_n\}$, *the soft nearest neighbor of* $l_i \in L$ *in the set* $M$ *is given by* $\tilde{m} = \sum_{j=1}^n \alpha_j m_j$, *where* $\alpha_j = \frac{e^{-d(l_i, m_j)/\tau}}{\sum_{k=1}^n e^{-d(l_i, m_k)/\tau}}$ *and* $\tau$ *is a temperature parameter.*

We first compute the soft nearest neighbor for $l_i \in L$ as $\tilde{m} = \sum_{j=1}^{n} \alpha_j m_j$. To satisfy the cyclic consistency constraint, $l_i$ should cycle back to itself. In order to enforce this, the nearest neighbor of $\tilde{m}$ should be $l_i$. In the differentiable formulation, we map the problem of cycling back to $l_i$ as a classification problem (Dwibedi et al., 2019). Given that we have $n$ points in $L$, we compute $n$ logits given by $o_j = -d(\tilde{m}, l_j), \forall j \in \{1, \ldots, n\}$ and let $\hat{s} = \text{softmax}(o)$. Let $s$ denote the 1-hot encoding where $s_j = 1$ and zero otherwise. Now the affinity cyclic consistency (ACC) constraint is enforced as the cross-entropy loss function $\sum_{j=1}^{n} -s_j log(\hat{s}_j)$. Note that the loss naturally prevents point collapse in embedding space as a trivial solution, because the softmax output compares the cycle back to the affinity of all points in $L$.

**Double augmentation:** In practice, real images can have many more active factors of variation than those which we desire to isolate. We introduce a simple modification to the above method in order to suppress nuisance factors of variation in images which are easily augmented. Instead of enforcing cycle consistency from an image $a_i \in A$ to its soft nearest neighbor in $B$ and then back to $a_i$, we allow the cycling back constraint to start from one augmentation of an image $a_i'$ and return back to another augmentation of the same image $a_i''$. By carefully selecting the augmentations $a_i'$ and $a_i''$ along certain active factors of variation, we can learn embeddings that are invariant to nuisance active factors of variation. We found the double augmentation to be critical in our pose transfer experiments involving real and synthetic cars in section 4.3.

## 4 EXPERIMENTS

We evaluate ACC's ability to isolate desired factors of variation in three settings: Shapes3D (Burgess & Kim, 2018), digit style isolation, and object 3D pose transfer from synthetic to real images.

### 4.1 SYSTEMATIC EVALUATIONS ON SHAPES3D

We quantitatively analyze ACC with the synthetic Shapes3D (Burgess & Kim, 2018) dataset, where we can freely control the active and inactive factors of variation in the two sets and measure quantities of interest in order to elucidate the inner workings of the method. Sample images from Shapes3D are shown in Figure 3 and consist of a geometric primitive with a floor and background wall. There are six factors of variation in the dataset: three color factors (wall hue, object hue and floor hue) and three geometric primitive factors (scale, shape and orientation). The full dataset consists of every possible combination of these six discrete generative factors.

In order to train ACC with a particular generative factor inactive, for each training step we randomly sample from among its possible values and hold it fixed across a set of inputs, while sampling uniformly across the remaining factors to generate a set of size 32. For example, Figure 3a shows two example sets with wall hue as an inactive factor. We train a network to embed the images into two dimensions and visualize one set of learned representations in Figure 3b; the mutual information between object and floor hues and the learned embedding is qualitatively evident. We measure the mutual information $I(U; G)$ (see Appendix B) between each of the generative factors and the embeddings, and repeat each experiment for 50 random seeds. In each subplot of Figure 3c, different factors are made inactive during training. We show the (nonzero) mutual information present even in the output from an untrained network, as well as the result of training without any set supervision. In this baseline, all generative factors are active during training and nothing is suppressed in the learned embeddings. Interestingly, without supervision, there arises a significant distinction between the hue factors and the others, presumably out of salience with respect to the network's capabilities.

The subsequent subplots in Figure 3c show the result of training with various generative factor(s) rendered inactive, indicated by a shaded box. Clear trends arise which allow a more precise definition of active factor isolation: information with respect to the inactive factors is noticeably suppressed while information about the remaining factors is enhanced in the learned representation. In other words, given weak set supervision over one factor of variation, ACC leads to learned representations which isolate factors of variation about which nothing was known beforehand. When all three hue factors are inactive we see scale, shape, and orientation feature most prominently in the learned representations, seemingly because the 'easy' hue factors have all been suppressed.

Additionally, we compare to settings where only set $A$ has supervision (inactive factors) and the other set $B$ consists of random samples over the entire dataset. Strikingly, there is no significant change in the information content of the generative factors in the learned representations, meaning ACC performs just as well in the much looser setting. Finally, in the bottom subplot of Figure 3c, we

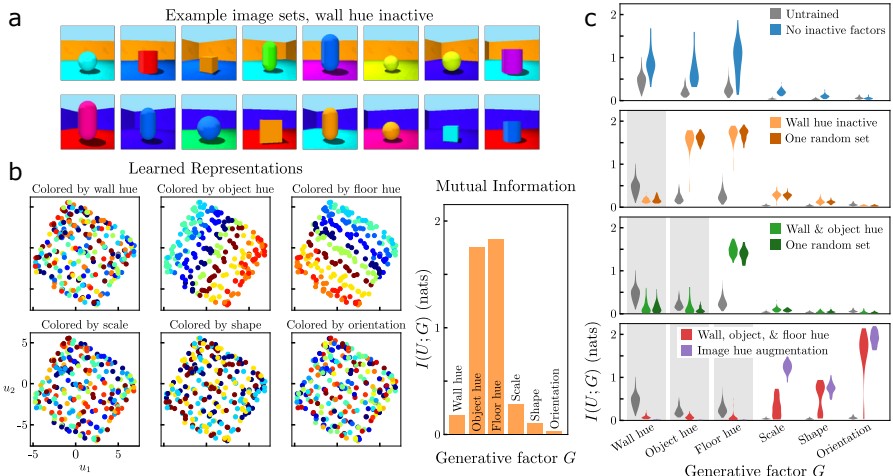

Figure 3: ***Analysis on Shapes3D.*** *(a) An example pair of training sets where wall hue is the only inactive factor in each set. (b) We train ACC on pairs of image sets as in (a) and embed to 2D. Each of the six plots displays the same 256 embeddings colored by their value for a different generative factor. The bar chart shows the mutual information between each of the factors and the learned representation. (c) We repeat the experiments with 50 random seeds and display the spread in mutual information values as a violin plot. Each subplot shows a different split of active/inactive factors during training. The output from an untrained network is shown in black. In the middle two subplots, we compare to the setting with one unconstrained input set, i.e. with no inactive factors of variation. The bottom subplot compares to a contrastive method which augments the overall hue of each image twice and uses the two versions as a positive pair. In each setting, ACC successfully suppresses information about inactive factor(s) of variation and enhances information about active factors.*

compare ACC results where all hue factors were inactive, to the case of hue double augmentation which suppresses all hue information from the learned embeddings. While the factor isolation effects are more pronounced in the latter, we emphasize that augmenting away nuisance factors is often not possible. This is the case in the middle two subplots, where augmenting hue would suppress all three hue factors indiscriminately and inhibit information with respect to the active variables. Thus ACC and double augmentation are complementary tools to operate on factors of variation in a dataset.

## 4.2 DIGIT STYLE ISOLATION

Handwritten digits, such as those from MNIST (LeCun & Cortes, 1998), have two main factors of variation: content and style. Here, content refers to the class of the digit (e.g., 2 or 8) and the rest of the factors of variation can be referred to as style (stroke width, orientation, writing style etc.). In this experiment, our aim is to learn embeddings that isolate digit style while being invariant to the digit class, with only set supervision on the digit class. We group images by class into sets of size 64 and embed to 8D using a convolutional network (See Appendix A for specifics). Images of the digit 9 are held out from training to probe the invariance of the learned embeddings to digit class.

Figure 4 (left) shows two-dimensional PCA plots of the learned embeddings next to those of un-trained embeddings. The PCA plots clearly indicate the stronger correlation between digit style (width, orientation etc.) and the learned embeddings in comparison to those of untrained embeddings. As further analysis, we use test digits from each of the 10 classes to retrieve the nearest neighbor digits in other classes in Figure 4 (right). We compare to the representations yielded by two VAE-based approaches which utilize grouped data to separate factors of variation: CC-VAE (Jha et al., 2018) in Figure 4 and ML-VAE (Bouchacourt et al., 2018) in Appendix F. Without having to learn a full description of the data, ACC yields style-correlated embeddings 100 times faster than the related generative approaches. This demonstrates the superior potential of ACC to isolate digit style without using any explicit supervision on styles.

## 4.3 OBJECT POSE TRANSFER FROM SYNTHETIC TO REAL IMAGES

We showcase the unique capabilities of ACC on the challenging task of 3D pose estimation of an object in an image. A common data setting, in which there is an abundance of synthetic data

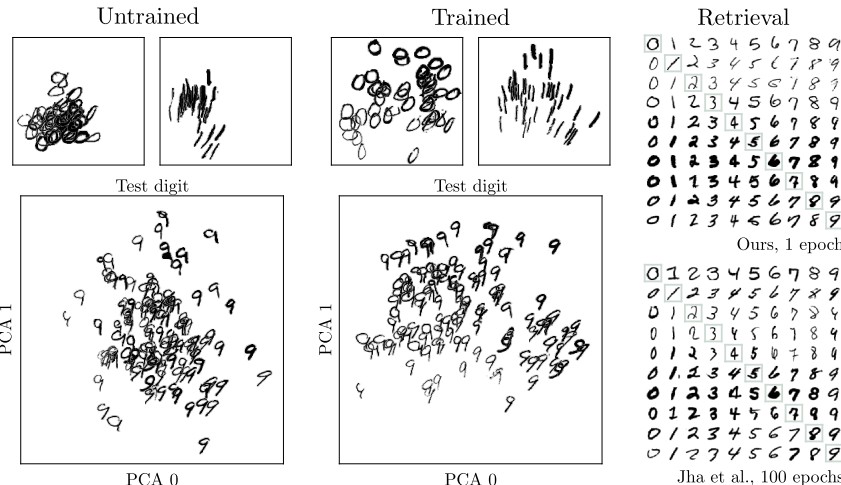

Figure 4: *Style isolation in MNIST. We define digit style as a combination of all factors of variation independent of the class. After training ACC with images grouped by digit, we evaluate the isolation of the style factors. We visualize embeddings from the test set using the top two PCA dimensions (accounting for more than 0.99 of the total variance in the trained embeddings). Before the network is trained, all the embeddings cluster together (top left), while after training (top middle) the embeddings fan out by style factors, primarily thickness and slant (this can be observed both within and across digits). The digit 9 is held out at training, yet embeddings of 9s (bottom middle) are similarly arranged by thickness and slant, showing ACC generalizes to unseen inactive factor values. On the right we show retrieval results, where the boxed images along the diagonal are queries and the other images in each row are the nearest embeddings for each digit class (all from the test set). ACC retrieves images closer in style than CC-VAE (Jha et al., 2018), a generative approach which also utilizes set supervision. Notably, ACC required 100x fewer training steps, highlighting a benefit of learning partial descriptions through discriminative approaches as compared to generative models.*

combined with unannotated real data, plays to the strengths of ACC. The method allows us to isolate pose information in learned representations by leveraging natural groupings of synthetic images where pose is the only active variable, even without any pose annotations at training. Additionally, an unconstrained second set provides a means to gradually incorporate unannotated real images which helps generalize object pose from the synthetic domain to the real.

We use the dataset included in KeypointNet (Suwajanakorn et al., 2018), which consists of renderings of ShapeNet (Chang et al., 2015) 3D models from viewpoints which are randomly distributed over the upper hemisphere. Set supervision is provided by grouping images according to their source 3D model (as in the upper image set of Figure 1). Other factors of variations such as object texture, lighting are also fixed, making viewpoint the only active factor within each set. We pair the synthetic images with real images from the CompCars (Yang et al., 2015b) and Cars196 (Krause et al., 2013) datasets for the car category, and 1000 images from the Pascal3d+ (Xiang et al., 2014) training split for chairs. All images are tight cropped and resized to 128x128.

In the first experiment, we discard pose annotations entirely and show that ACC yields representations which are pose informative with respect to real images, solely using groupings of synthetic images by particular model. We train with set $A$ purely synthetic and grouped by ShapeNet model, and set $B$ unconstrained. For the first 10k iterations set $B$ is synthetic images randomly sampled across all models and viewpoints, and for the following 10k iterations real images are factored in as 5.5% of each set $B$. We found it beneficial to suppress nuisance active factors of variation in the images, such as the precise position of the object in the frame, by optimizing the double augmentation loss explained in Section 3. Each image is randomly augmented twice with a combination of cropping, recoloring, and edge enhancement via Sobel filter. The network adds a few layers to the base of an ImageNet-pre-trained ResNet (He et al., 2015) before embedding to 64 dimensions (specifics in Appendix A). We find cosine similarity with $\tau = 0.1$ outperforms L2 distance.

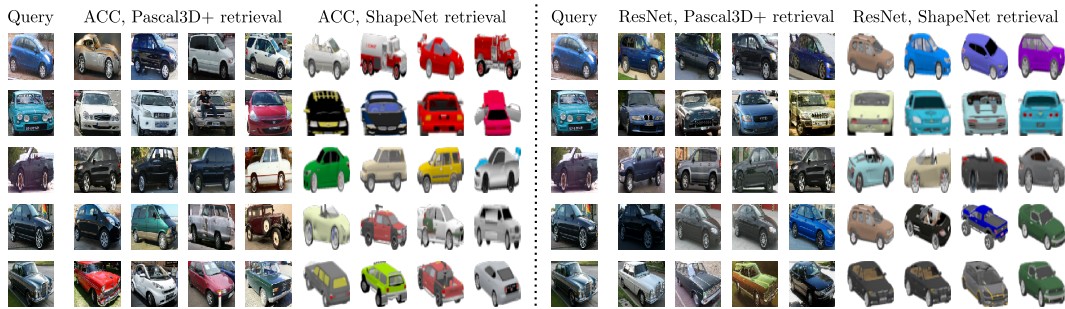

Figure 5: *Retrieval results from ACC and ResNet embeddings. For each query image from the Pascal3D+ test split, we display the four nearest neighbors in embedding space, out of 3200, from the Pascal3D+ train split and the ShapeNet images. Note how ACC yields similar representations for images which are often visually quite different, in contrast to the ResNet output. This serves as qualitative evidence that pose is being effectively isolated in the ACC-trained embeddings.*

We evaluate the learned representations on the images in Pascal3D+ by using nearest neighbor lookup between embedded test images and a dictionary of 1800 synthetic images with ground-truth pose. We compare to the 16,384-dimensional output from the ResNet base network and to embeddings learned with the VAE-based approaches of Jha et al. (2018) and Bouchacourt et al. (2018). Quantitative results are shown in Table 1 and retrieval examples in Figure 5. We additionally report a flip invariant metric to highlight how well ACC performs modulo flip symmetry, a particular difficulty for cars. We consistently outperform the baselines with the gap significant in many cases, especially for chairs, showing the efficacy of ACC. The significant difference between ACC and the generative approaches underscores the importance of meaningfully incorporating unannotated real images during training; there is no simple means to do so with either VAE-based method.

In the second experiment (Table 2), we make use of the pose annotations for the synthetic images by incorporating ACC into the spherical regression framework of (Liao et al., 2019). Specifically, we add a small spherical regression head after the ACC-conditioned representations (Figure 6) and train on a weighted sum of the two losses. Even without any real images during training, ACC improves performance, presumably by better conditioning the intermediate latent space. A significant boost to performance results when a small amount of real images (2%) are titrated in gradually over training, for both object categories. See Appendix E for ablative studies.

| | Cars | | | | Chairs | | | |
| | | | Flip invariant: $\min(\theta, 180° - \theta)$ | | | | Flip invariant: $\min(\theta, 180° - \theta)$ | |
| | Med Err (°) ↓ | Acc@30° ↑ | Med Err (°) ↓ | Acc@30° ↑ | Med Err (°) ↓ | Acc@30° ↑ | Med Err (°) ↓ | Acc@30° ↑ |
|---|---|---|---|---|---|---|---|---|
| ResNet (pre-trained) | 16.0 | **0.65** | 10.1 | 0.93 | 71.0 | 0.30 | 32.5 | 0.46 |
| CCVAE (Jha 2018) | 54.8 | 0.26 | 30.4 | 0.48 | 79.5 | 0.19 | 41.3 | 0.35 |
| ML-VAE (Bouchacourt 2018) | 75.6 | 0.27 | 29.8 | 0.50 | 87.2 | 0.16 | 44.5 | 0.33 |
| ACC (Ours) | **14.1** | **0.65** | **8.4** | **0.95** | **35.1** | **0.47** | **25.7** | **0.55** |

Table 1: *Pose estimation without any pose annotations during training. Median error and accuracy metrics on Pascal3D+ car and chair test datasets. We obtain the pose with nearest neighbor lookup into 1800 synthetic images with GT pose, using different embeddings shown in different rows. ACC outperforms the VAE-based approaches and the high dimensional ResNet embeddings.*

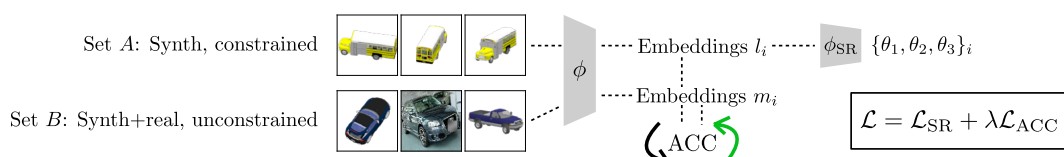

Figure 6: *Strengthening spherical regression with ACC. We append a spherical regression head (Liao et al., 2019) to the network and find that an ACC loss on the intermediate embeddings significantly improves performance.*

| | Cars | | Chairs | |
|---|---|---|---|---|
| | Med Err (°) ↓ | Acc@30° ↑ | Med Err (°) ↓ | Acc@30° ↑ |
| Liao et al. (2019) | 12.3 | 0.85 | 30.8 | 0.49 |
| + ACC | 11.0 | 0.79 | 28.1 | 0.52 |
| + ACC (2% unannotated real) | **9.3** | **0.87** | **26.0** | **0.55** |

Table 2: ***Performance boost to spherical regression by incorporating ACC.*** *We show the effectiveness of incorporating ACC as an additional loss term when the data consists of annotated synthetic images and unannotated real images. ACC provides a means to incorporate the latter which significantly helps bridge the domain gap from synthetic to real pose estimation.*

## 5 DISCUSSION

Leveraging only data groupings, ACC produces informative representations with respect to factors of variation disjoint from the subset of factors for which there is weak supervision. The loss is optimized when approximate correspondence can consistently be found between all inputs of set $A$ and all inputs of set $B$, for all pairings of $A$ and $B$. The inactive factor(s) are common to all elements of a set and therefore offer no distinguishing information to help with correspondence; thus they are left out of the representation. One set can be unconstrained because only active factors present in both sets can be used to find correspondence, meaning the more constrained of the two sets dictates which factors are extracted. For the example sets of Figure 1, the color of the car cannot be used to find correspondence because it does not distinguish between the elements of set $A$.

It is evident in the mutual information measurements of Figure 3 that only a subset of the active factors of variation are present in the learned representations. This can be partly attributed to the low dimensionality of the embeddings – a design choice to facilitate the measurement of mutual information, which is notoriously problematic in higher dimensions – though we show in Appendix B that the effect is also present for 64-dimensional embeddings. A correspondence can be made when each element is embedded according to only a single active factor of variation common to both sets. This was the case for Dwibedi et al. (2019), where the progression of an action (e.g., bowling) was the only active factor of variation (with scene specifics being inactive, fixed per video). We show that ACC extends naturally to data with multiple active factors of variation. This seems to arise from randomness in the input sets: on average the loss is decreased by embedding random points to more independent dimensions (Appendix D). In practice, factors differ in salience. The hue-related generative factors of Shapes3D are easier for the specific network to extract, and training effectively ceases once a correspondence utilizing these factors is found. Similarly, nuisance factors of variation in the images of cars and chairs are easier to extract than pose, which is why double augmentation helped to encourage the network to isolate pose.

The task of finding correspondence is naturally suited to domain transfer, as showcased in the pose estimation experiments of Section 4.3. ACC provides a means to incorporate unannotated images from a similar domain, as the loss incentivizes overlooking factors of variation which do not aid in the correspondence task. We found the fraction of images incorporated should remain small, presumably because it is possible to use one factor to embed images from one domain and another for a second domain, as long as the representations are co-located in embedding space.

The breadth of the three scenarios explored in this paper showcase the generality of ACC, and there is no reason the method should be restricted to image data. The cycle consistency loss operates on embeddings and is thus insensitive to the modality of the input data. Additionally, while we experimented with two forms of the distance metric in embedding space – L2 and cosine similarity – nothing about the loss necessitates these, and presumably in some scenarios employing other distance metrics or structures of embedding space would be advantageous.

## 6 CONCLUSION

In this work, we show how affinity cycle consistency can be utilized to isolate factors of variation provided only weak set supervision. Through extensive experiments on synthetic and real data, we show the technique can be applied in a number of different set supervision scenarios, including one where only a subset of the data requires any set supervision at all. This case is particularly important as it allows training with unsupervised real image data, and we validate this with promising experiments on the challenging problem of isolating 3D object pose in real images.

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

## A IMPLEMENTATION DETAILS

For all experiments we use the ADAM optimizer ($\beta_1 = 0.9$, $\beta_2$=0.999). Padding for convolutional layers is always 'valid.'

### A.1 SHAPES3D

For the experiments of Figure 3 we trained for 2000 steps with a learning rate of $3 \times 10^{-5}$. We used a stack size of 32 and squared L2 distance as the embedding space metric, with a temperature of 1.

| Layer | Units | Kernel size | Activation | Stride |
|---|---|---|---|---|
| Conv2D | 32 | 3x3 | ReLU | 1 |
| Conv2D | 32 | 3x3 | ReLU | 1 |
| Conv2D | 64 | 3x3 | ReLU | 2 |
| Conv2D | 64 | 3x3 | ReLU | 1 |
| Conv2D | 128 | 3x3 | ReLU | 1 |
| Conv2D | 128 | 3x3 | ReLU | 2 |
| Flatten | – | – | – | – |
| Dense | 128 | – | ReLU | – |
| Dense | Embedding dimension (2) | – | Linear | – |

Table 3: **Architecture used for Shapes3D experiments (Section 4.1).** Input shape is [64, 64, 3].

| Layer | Units | Kernel size | Activation | Stride |
|---|---|---|---|---|
| Conv2D | 32 | 3x3 | ReLU | 1 |
| Conv2D | 32 | 3x3 | ReLU | 1 |
| Conv2D | 32 | 3x3 | ReLU | 2 |
| Conv2D | 32 | 3x3 | ReLU | 1 |
| Conv2D | 32 | 3x3 | ReLU | 1 |
| Flatten | – | – | – | – |
| Dense | 128 | – | ReLU | – |
| Dense | Embedding dimension (8) | – | Linear | – |

Table 4: **Architecture used for MNIST experiments (Section 4.2).** Input shape is [28, 28, 1].

## A.2   MNIST

For the MNIST experiments, the stack size is 64. We use a learning rate of $10^{-4}$ and train for 1000 steps. We used squared L2 distance as the embedding space metric and a temperature of 1, though as long as the length scale set by the temperature is larger than the initial point spread from the randomly initialized network, it does not seem to matter.

## A.3   POSE ESTIMATION

| Layer | Units | Kernel size | Activation | Stride |
|---|---|---|---|---|
| ResNet50, up to conv4_block6 | – | – | – | – |
| Conv2D | 256 | 3x3 | ReLU | 1 |
| Global Average Pooling | – | – | – | – |
| Flatten | – | – | – | – |
| Dense | 128 | – | tanh | – |
| Dense | Embedding dimension (64) | – | Linear | – |

Table 5: **Architecture used for pose estimation experiments (Section 4.3).** Input shape is [128, 128, 3].

For both the pose estimation lookup and regression tasks, we use the same base network to embed the images. For regression, the embeddings are then fed, separately for each Euler angle, as input to a 128 unit dense layer with tanh activation, which is then split off into two dense layers with 2 and 4 units and linear activation for the angle magnitude and quadrant, respectively, as in (Liao et al., 2019). The angle magnitudes are passed through a spherical exponential activation function Liao et al. (2019), which is the square root of a softmax. The magnitudes are then compared with ground truth $(|\sin\phi_i|, |\cos\phi_i|)$, with $i$ spanning the three Euler angles, through a cosine similarity loss. The quadrant outputs are trained as a classification task with categorical cross entropy against the ground truth angle quadrants, defined as $(\text{sign}(\sin\phi_i), \text{sign}(\cos\phi_i))$.

For the lookup task, the network trained for 20k steps, with the first half of training purely synthetic images, and then the second half with 5.5% real images folded into the unconstrained stack. For spherical regression, training proceeds for 60k steps with a learning rate that starts at $10^{-4}$ and

decays by a factor of 2 every 20k steps. Each minibatch consists of 4 pairs of image sets, each of size 32. We use cosine similarity and a temperature of 0.1 for lookup and 0.05 for regression. To maintain consistency between how the embeddings are processed for the ACC loss and how they are fed into the regression sub-network, the embeddings are L2-normalized to lie on the 64-dimensional unit sphere before the regression.

To more closely match the distribution of camera pose in real images, we filter the ShapeNet renderings by elevation: 0.5 radians and 1.3 radians for the max elevation for cars and chairs, respectively.

## B MUTUAL INFORMATION CALCULATION AND SHAPES3D IN HIGHER DIMENSIONS

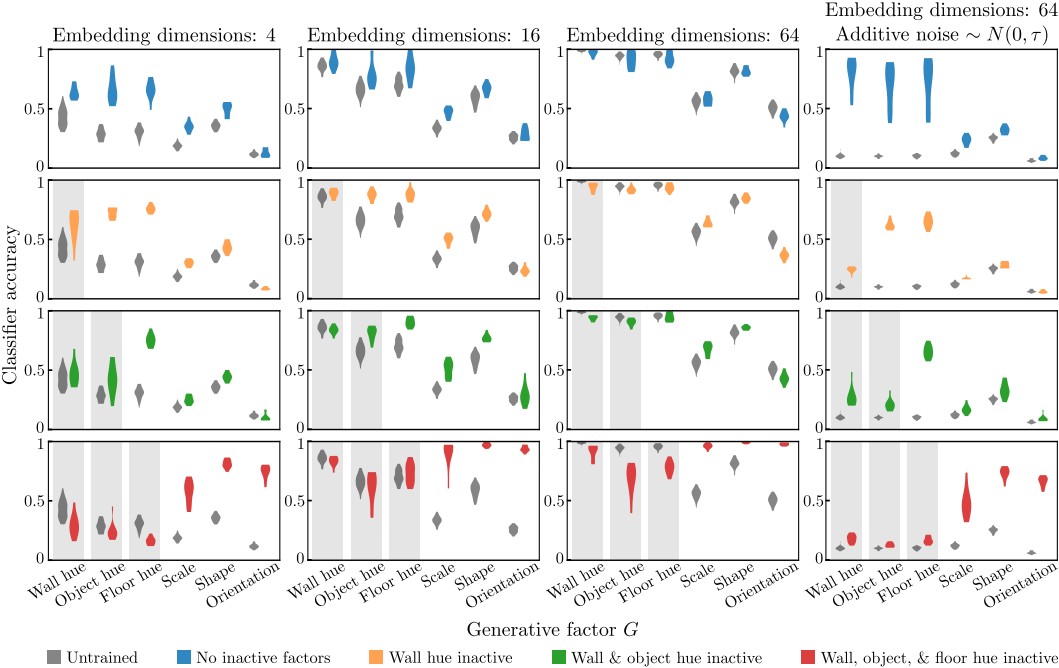

Figure 7: ***Probing information content in higher dimensions via classification.*** *We repeat the experiments of Section 4.1 in 4, 16, and 64 dimensional embedding space, though with 10 replicas each instead of 50. As a proxy for the mutual information, we use the test set classification accuracy of simple fully connected networks trained to classify each of the six generative factors. As before, in each subplot we display in gray the baseline results from embedding with an untrained, randomly initialized network. Also as before, the colors of each subplot indicate the same information as the shaded columns: which of the generative factors were inactive while training ACC. In the rightmost subplots, Gaussian-distributed random noise was added to the embeddings to effectively remove information on length scales less than the characteristic length scale of the ACC loss, the square root of the temperature.*

For the experiments of Section 4.1, we used an embedding dimension of 2 in order to facilitate measuring the mutual information. We approximated the 2D distribution by embedding all $N = 480,000$ images of the Shapes3D dataset and using a histogram with $\sqrt{N}$ bins. The same bins were then used for all of the various conditional distributions, where the embedding distribution conditioned on each possible value of each of the six generative factors was evaluated. The values of entropy calculated from this simple method were found to be insensitive to the number of bins, within a range, and more reliable than the popular non-parametric approach of (Kraskov et al., 2004).

It is fair to question whether the low dimensionality of the embeddings affects the behavior of ACC. We repeat the experiments of Section 4.1 with higher dimensional latent spaces and probe the information content by training a simple classifier (3 layers with 32 units each, ReLU activation), using the learned representations as input, for each of the generative factors.

The test set classification accuracies, shown in Figure 7, have many noteworthy quirks. As a baseline, the embeddings output by a randomly initialized (untrained) network are, sensibly, more successfully classified as the number of dimensions increases. The isolation of active factors that occurs for two dimensional embeddings (Figure 3c) is more subtle in higher dimensions. As in 2D, a subset of the active factors feature more prominently in the learned representations, and that subset is more or less the same: unless all three hue factors are inactive, the three geometric factors (scale, shape and orientation) are hardly affected. When all three hue factors are inactive, information with respect to all three geometric factors is clearly enhanced, regardless of the dimension.

Because even random embeddings are easily parsed by a classifier in higher dimensions, we do not see the obvious suppression of inactive factors as in two dimensions. This is reasonable, however, given that the ACC loss operates over a characteristic length scale, set by the temperature parameter $\tau$ in both the soft nearest neighbor calculation and the classification loss. In other words, two embeddings separated by much less than this length scale effectively have a separation of zero in the eyes of the loss, and there is no incentive to further collapse them. As information is about the capacity to infer the value of one random variable given another, it is only destroyed in the case where multiple inputs map to the same output. This depends on the granularity of observation, as what qualifies as the *same* is different for float precision versus a hyperopic ACC loss. Thus when the ACC training leads to embedding separations with respect to a particular generative factor which are much less than the characteristic length scale, the information content has been removed from the perspective of the loss.

To be specific, when using L2 (Euclidean) distance as the similarity metric, the temperature $\tau$ is the characteristic length scale. When using L2 squared distance, as in the MNIST and Shapes3D experiments, the square root of the temperature is the characteristic length scale.

We expect, then, that the learned embeddings in 64 dimensions contain information about the active factors on length scales greater than $\sqrt{\tau}$ and about the inactive factors on length scales less than this. A simple test is to introduce random noise into embedding space, removing information on length scales less than that of the noise. We add Gaussian-distributed noise with variance $\tau$ to the embeddings during the training of the classifier, and show the resulting test set classification accuracies (without noise) in the rightmost plot of Figure 7. The untrained network results are not very informative because the length scale of the embeddings is whatever resulted from the randomly initialized weights of the embedding network, which happened to be less than $\sqrt{\tau}$. The accuracies all hover around $1/n$, with $n$ the number of possible values for generative factor: random guessing, in other words. The classification accuracies for the learned representations, however, now display the same behavior as did the mutual information in two dimensions. The inactive factors are suppressed, and the same active factors are enhanced in each active-inactive factor scenario.

While not so clean as the mutual information measurements, the classification accuracies help generalize the notion of active factor isolation: the ACC loss organizes embedding space around a subset of active factors that permit a correspondence between input sets, and information with respect to inactive factors is destroyed to the degree that the ACC loss can do so.

## C  FOUR INACTIVE FACTORS FOR SHAPES3D: ALL HUE PLUS A GEOMETRIC FACTOR

We expand upon the results of Figure 3 by training with more constrained input sets. We show in Figure 8 results from experiments with four inactive factors: the three hue-related factors and then one of the geometric factors. The aggregate behavior essentially mirrors that of the bottom subplot of Figure 3c where the three hue factors are inactive, but where the additional inactive factor is suppressed.

A noteworthy behavior is apparent when visualizing the embeddings directly for individual runs, as in Figure 8a, The shape generative factor, when active, is consistently partitioned in embedding space into only two groups. The four shapes are cube, cylinder, sphere, and pill; if embedding space were partitioned perfectly by shape the mutual information would be the natural log of 4, to which it never gets particularly close. The colocated embeddings are cubes with cylinders and spheres with pills. Evidently whether the top of the shape is round or flat is more salient than the other details, providing another example where the salience of factors affects their isolation in the learned embeddings.

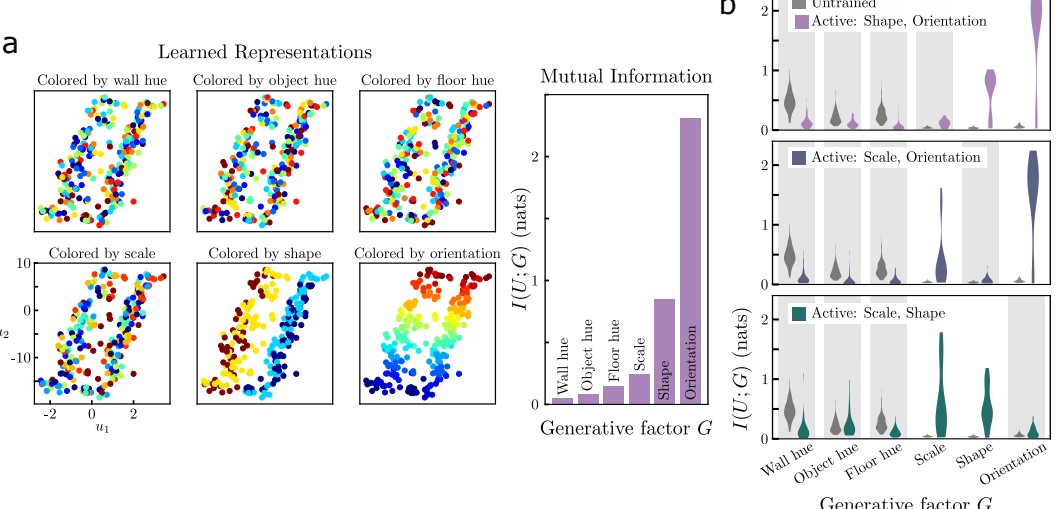

Figure 8: ***Highly constrained inputs for Shapes3D experiments.*** *We extend the experiments of Section [4.1](#) by constraining the input sets with one additional generative factor, to better probe the difference between the three hue and the three geometric factors. In **(a)** we show one example of learned representations where the three hue factors and scale are inactive factors in each training set. Interestingly, the shape factor – of which there are four possible values – seems to be split into two groups, one with cylinders and cubes and the other with pills and spheres. We observed this particular grouping to happen in the majority of the cases for this active-inactive split, indicating another level of salience difference with respect to the embedding network. In **(b)** we measure the mutual information $I(U;G)$ with respect to all six generative factors, as in Figure [3](#)c, where the fourth (geometric) inactive factor is the scale, shape, and orientation, respectively.*

## D    MULTIPLE FACTORS OF VARIATION AND THE EFFECT OF SET SIZE

If correspondence between two sets can be found with only a single factor of variation, why do the experiments of this paper suggest ACC isolates multiple factors of variation? To be specific, in almost all of the Shapes3D experiments, multiple generative factors were present in the learned representations. Presumably a correspondence between MNIST digits could be found using stroke thickness, yet the embeddings always contain slant information as well. In the pose experiments, only embedding azimuth would suffice to allow a correspondence between images, yet elevation information was also clearly present.

In Figure [9](#) we run a simple Monte Carlo experiment where two sets of points are randomly sampled from a uniform distribution, representing ideal embeddings whose factors of variation are randomly distributed from the same distribution, and the value for the ACC loss is evaluated. The loss is averaged over 10,000 random draws. In the normal setting where the distribution in embedding space is learned via training, temperature has little effect because the distribution can be expanded or contracted to best fit the length scale set by temperature. In this simulation, the distribution is fixed so the temperature which optimizes the loss needs to be found. We observed that the length scale set by the temperature roughly scales with the average inter-point distance, but not exactly, so the value was optimized numerically.

Interestingly, when the number of points in each set is larger than O(10), the ACC loss can be lowered by increasing the dimension of the distribution. In other words, there is an incentive in the loss to find multiple independent factors of variation. Additionally, the effect grows as the set size grows.

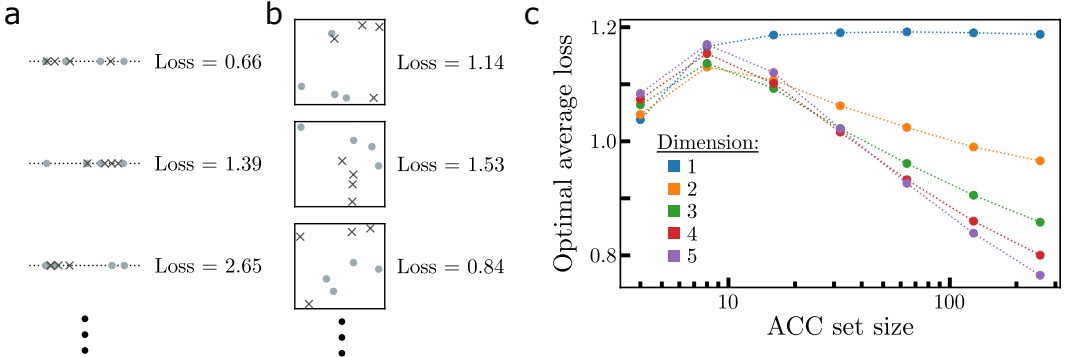

Figure 9: ***The case for finding more than one factor of variation, through a simple example.*** *We model the embeddings that would be learned from randomly distributed factors of variation as points sampled uniformly over the unit interval in one to five dimensions. **(a)** Displayed are three random draws, with set size equal to 4 and dimension 1, and with corresponding loss values for the numerically optimized temperature T=0.05. See Appendix text for why temperature needs to be optimized in this simulation but not in general. The × and circle markers designate Set A and B. **(b)** Same, but for dimension 2 and numerically optimized temperature T=0.11. **(c)** The ACC loss averaged over 10,000 random draws, for varying set size and dimension of the uniform distribution. While mapping points from two sets along one dimension allows a correspondence to be found, we see that in the presence of stochasticity, multiple independent dimensions lead to lower average values for the loss when the set size is larger than O(10). We take this as suggestive for why ACC pulls out more than just a single factor of variation, when the factors are of similar salience.*

# E  POSE ESTIMATION ABLATIVE STUDIES

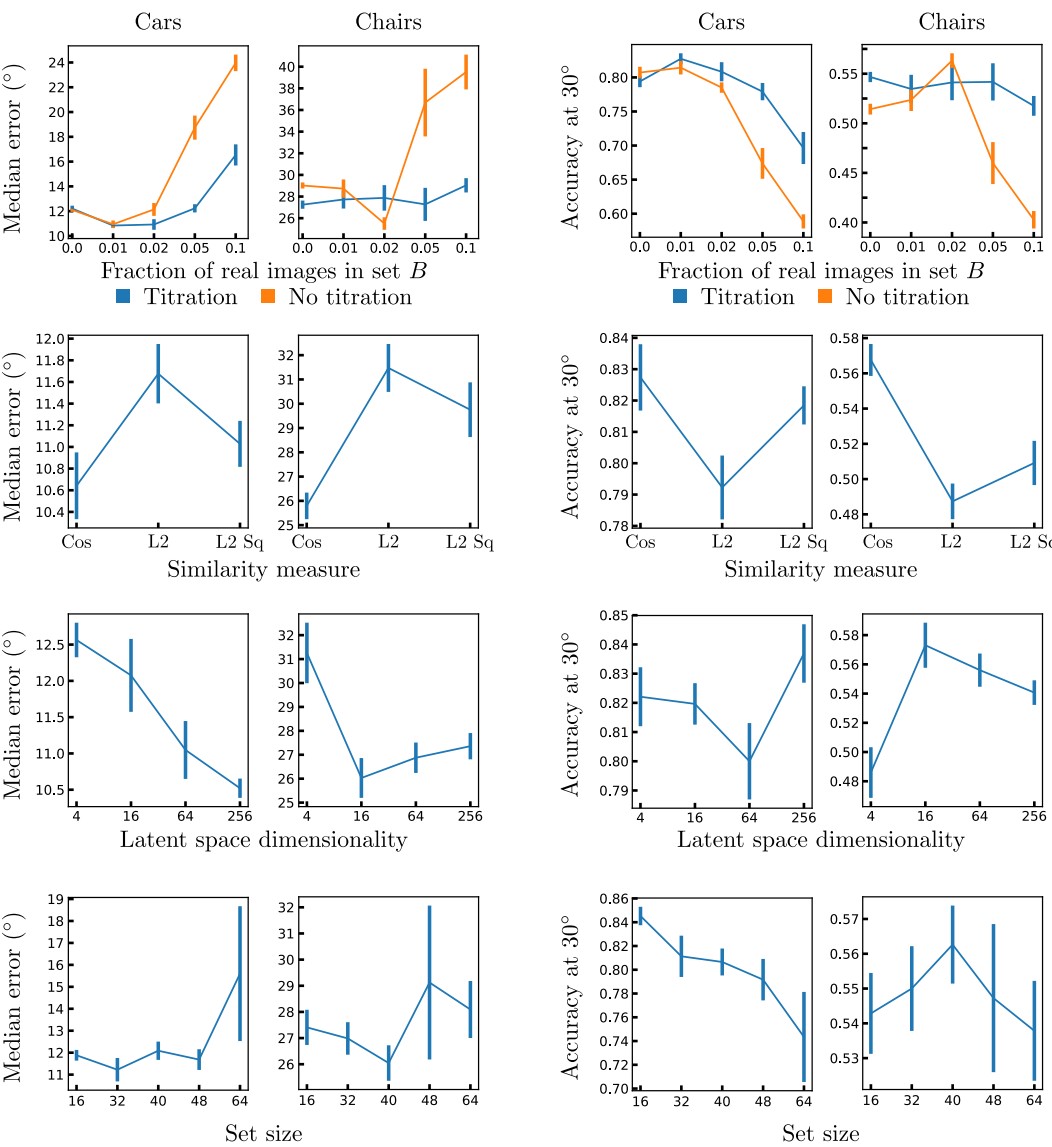

Figure 10: ***Ablative studies with spherical regression + ACC network.*** *Error bars are the standard error of the mean over 10 random seeds for each configuration, with less than 1% of the runs discarded for lack of convergence. We show results on the Pascal3D+ test split for the car and chair categories. For each row, the training configuration is the same as described in Appendix A with only the listed aspect of training being changed. In the first row, no titration means to the fraction of real images in set B are present from the beginning of training. The three similarity measures in the second row are cosine similarity, L2 (Euclidean) distance, and squared L2 distance.*

# F EXTENDED DIGIT STYLE ISOLATION RESULTS

Figure 11: **Retrieval results over the course of training, comparison.** *We compare retrieval on the test set of MNIST at various stages of training ACC and the two VAE-based approaches mentioned in the main text. As in Figure 4, the query images are the boxed images along the diagonal, and each row is the nearest representative for each class in embedding space. Also as before, in all cases the digit 9 was withheld during training.*

We compare digit style isolation on MNIST using the output of ACC and the style part of the latent representations yielded by the VAE-based approaches of Jha et al. (2018) and Bouchacourt et al. (2018). Interestingly, ML-VAE appears to embed the digits with respect to stroke thickness and slant very similarly to ACC at the beginning of training, long before any realistic images are able to be generated, but this clear interpretability of the embeddings fades as training progresses.

