# OpenReview forum: "Leveraging affinity cycle consistency to isolate factors of variation in learned representations"
_ICLR.cc/2021/Conference — Reject_

### Official Review · AnonReviewer1 · 2020-10-26

**Rating:** 6
**Confidence:** 3

**Review:**

This paper uses affinity cycle consistency to isolate factors of variation with only weak supervision on set membership. It has extensive experiments with both synthetic and real data.

The strength is that the problem setting is reasonable and important. The algorithm is sounding, and the evaluation is valid. The weakness is that the experiments and discussions do not cover enough cases for extracting and suppressing factors.

I recommend that this paper is marginally above the border.

The positive reasons are the following. Isolating factors is a difficult and important problem, so the progress is good. Using weak supervision is practically convenient. Also, the paper shows that it helps a synthetic-to-real transfer problem. The following are some concerns.
1. Extracting all active factors may need more samples in set A.
In the experiment of Figure 3c, second from the top, the learned representation does not isolate active geometric factors, indicating it does not include the information. This means set A does not contain enough variety of elements so that the object and floor hues are enough to distinguish the elements. To isolate all the active factors, set A should contain more elements than used in the experiment. It would be more helpful to find how many elements are necessary from theoretical and / or empirical perspectives.
2. Suppression of complicated factors.
The paper does not tell why the inactive factors can be suppressed for unseen values, especially when it is inactive in both set A and B. When inactive, a factor takes a fixed value. So when it is inactive both in A and B, the factor has at most two values in training. However, in the test, it is suppressed for all (or most) possible values. The generalization capability from two values to many values requires more explanations, especially when the factor is complicated (e.g. non-linear) to extract. From experiment perspective, Figure 3c covers the cases of inactive ‘easy’ ’hue factors. It would be helpful to see what happens when the three hue factors and a geometric factor (e.g. orientation) are all inactive.

Additional feedback:
The last section is named “Discussion”, but it looks like “Conclusion” from its contents and position.
The paper does not provide code to reproduce the results.
Some figures and letters on them are very small.
Grammars:
“a affinity cycle consistency loss”
“for a each generative factor”

---

> ### Author Response · Authors · 2020-11-20
> **More material about factor of variation isolation by ACC, and reproducibility**
>
> We are glad the Reviewer appreciates the problem as “difficult and important” and thinks “progress is good”.  We follow the reviewer’s suggestions and include more material about what factors are isolated and why, as well as implementation details.
>
> *The experiments and discussions do not cover enough cases for extracting and suppressing factors.*
>
> We have greatly expanded discussion and experiments around extraction and suppression of factors of variation.
> - Rewrote Discussion Sec.
> - Added Appx. B about what factor isolation means in higher dimensions by way of the Shapes3D experiments
> - Added Appx. C with more constrained Shapes3D experiments (more inactive factors)
> - Added Appx. D with Monte Carlo results that show why the task of finding correspondence is benefitted by multiple independent factors
>
>
> *It would be more helpful to find how many elements are necessary from theoretical and / or empirical perspectives.*
>
> This is a good observation -- that in order to optimize the loss and find correspondence between set A and set B, the size of the two sets should influence how many factors the representations extract.
> - Added Appx. C with Monte Carlo results which show that using larger sets encourages finding more independent factors of variation.
>
> *Suppression of complicated factors… When inactive, a factor takes a fixed value. So when it is inactive both in A and B, the factor has at most two values in training…*
>
> To clarify, when a factor is inactive, it takes a fixed value only for a single input set.  The next training sets resample the inactive factor anew from among its possible values, so over the course of training the inactive factors are well sampled.
> - Clarified text of Sec. 4.1: “In order to train ACC with a particular generative factor inactive, **for each training step** we randomly sample from among its possible values and hold it fixed across a set of inputs...”
>
> *It would be helpful to see what happens when the three hue factors and a geometric factor (e.g. orientation) are all inactive.*
>
> We agree, and added a new Appx. with these results.  Interestingly another instance arises where the factors of variation have different salience to the embedding network: while there are four possible shapes, the embeddings consistently split into only two groupings.  Evidently distinguishing between the cylinder and cube, and the sphere and pill, respectively, are harder than distinguishing whether the top of the primitive is rounded or flat.
> - Added Appx. C, which repeats the experiments of Fig. 3c but with all possible inactive factor combinations of three hue factors and one geometric factor.
>
> *The paper does not provide code to reproduce the results.*
>
> - Included iPython notebook to reproduce digit style isolation results on MNIST (supplemental material)
> - Added Appx. A with implementation specifics for all experiments
> - We plan to post the code for the Shapes3D and pose estimation experiments at a later date
>
> *The last Sec. is named “Discussion”, but it looks like “Conclusion” from its contents and position.*
>
> We have added a new Discussion Sec. to consolidate various insights from the experiments, and retitled the prior Discussion to be the Conclusion.
>
> *Some figures and letters on them are very small.*
>
> We revised Fig. 5 to improve its readability.

---

> > ### Comment · AnonReviewer1 · 2020-11-23
> > **Further question**
> >
> > Thank you for answering the questions and updating the submission. I have a question regarding the following sentence.
> >
> > “In order to train ACC with a particular generative factor inactive, for each training step we randomly sample from among its possible values and hold it fixed across a set of inputs, while sampling uniformly across the remaining factors to generate a set of size 32.”
> >
> > It seems in this setting, the distribution of other factors is not influenced by the selected value of the inactive factor. If so, this means, statistically, the inactive factor is marginally independent of other factors: P(inactive, other) = P(inactive) P(other). Is this an implicit constraint of the algorithm, in addition to the set membership?

---

> > > ### Author Response · Authors · 2020-11-23
> > > **Marginal independence is the goal, not a constraint**
> > >
> > > Thank you for the insightful question.  No, we do not assume an independence constraint between different factors of variation.
> > >
> > > Instead of being an implicit constraint, marginally independent factors are the factors of variation sought by the loss.  A simple example from the MNIST dataset: a factor of variation which exists only for 2s is whether the bottom has a loop or not.  This factor of variation is dependent on the inactive factor (digit class), and would help with distinguishing from among a set of images of the digit 2, but does not help in finding correspondence with a set of images of the digit 3.
> > >
> > > For the pose estimation experiments, there are many factors of variation dependent on the specific car model, e.g. the pixel distance between the two headlights in the image.  Some of these factors may be the start to finding pose but their dependence on the inactive factor leads to suboptimal performance with respect to the ACC loss.  Pose is a higher level factor of variation which can be used throughout training for the correspondence task precisely due to its marginal independence.
> > >
> > > The simplicity of the Shapes3D dataset allows us to fully enumerate and factorize the factors of variation, making the analysis easier but perhaps giving the impression that the method requires such cleanly separable factors of variation.

---

### Official Review · AnonReviewer4 · 2020-10-27
**This paper applies a weakly-supervised Affinity Cycle Consistency loss to recognize factors of variation in image data sets and learn the image embedding representation of each identified factor.**

**Rating:** 3
**Confidence:** 4

**Review:**

This paper applies a weakly-supervised learning approach to identify factors of object postures in an image dataset. The core idea is to introduce two sets of images. The first set is the reference data set with grouped objects of different active/inactive posture constraints. This set is used to provide weak supervision information in posture identification. The second set is the probe set. It does not necessarily require posture grouping of objects. Affinity Cycle Consistency loss is set up to automatically map objects of similar active postures between the two image sets (objects of similar postures are supposed to be the nearest neighbors in the learned embedding space). The experimental study verifies the validity of the proposed factor isolation algorithm.

Generally this paper is well written and clearly explains the motivation/problem definition. However, we have the following concerns on the innovative contribution of this work.

The innovation of this paper is very limited. The core technology applied in this work was originally proposed by Dwibedi et al in the paper "Temporal Cycle-Consistency Learning".  As cited in Section.1 of this paper, this work employs directly the Cycle-Consistency learning mechanism (while in a different application scenario). The only difference is: this work is built based on affinity relation between pairs of images with similar postures, while the original idea was applied for temporal sequence alignment in video processing. Not significant algorithmic innovation is introduced, compared to the previous work.  It is clearly below the threshold for a high-quality venue like ICLR.

---

> ### Author Response · Authors · 2020-11-20
> **Emphasizing the generality of the method**
>
> Reviewer 4 is concerned about the ‘innovative contribution’ of our work, with the essence of the criticism contained the following sentence:
>
> *The only difference is: this work is built based on affinity relation between pairs of images with similar postures, while the original idea was applied for temporal sequence alignment in video processing.*
>
> We respectfully disagree.
>
> First, the central premise of the paper is the generality of the method, with pose estimation one of three use cases intended to showcase its breadth of applicability.  The pose estimation of Sec. 4.3 serves to elucidate novel aspects of ACC -- such as the ability to have an unconstrained second set -- and ground its representation-learning capabilities in a challenging and realistic problem setting.  The paper, as a whole, is about elevating a narrow-purpose method to temporally align video sequences to a significantly broader range of applications.  We demonstrate that ACC is a powerful discriminative approach to interrogating datasets, and take important steps to understanding its capabilities, limitations, and realm of application.
>
> Second, we introduce multiple expansions to the original method of Dwibedi et al. which further serve to generalize the method.  We show that one of the two input sets during training can be completely unconstrained, which allows us to incorporate unannotated, out-of-domain images during the pose estimation task.  In a set of pose estimation experiments added in the resubmission to Sec. 4.3, we demonstrate that the ability to incorporate unannotated real images significantly improves pose regression on real images over the spherical regression framework of Liao et al. (2019).  Additionally, we modify the cycle consistency loss to allow extra control over nuisance factors of variation by way of double augmentation, and measure the efficacy in Fig. 3c.
>
> Ultimately, the modifications to the original method of Dwibedi et al. further serve to generalize it and open the door for a far wider range of applications than aligning videos.

---

### Official Review · AnonReviewer3 · 2020-10-28
**Leveraging affinity cycle consistency to isolate factors of variation in learned representations**

**Rating:** 4
**Confidence:** 4

**Review:**

The paper presents an approach to isolate factors of variation using weak supervision in the form of group labels. The proposed method Affinity Cycle Consistency (ACC)  claims to work with these group labels, which are weaker than the more common, one factor per group type labeling.  An important aspect of this approach is that it does not attempt to disentangle the factors of variation, but only capture (or isolate) them in the latent space.

Following are the strengths of the paper:
+ It uses a very simple strategy of combining soft nearest neighbors with cycle consistency in the latent space to achieve the ability of isolating factors of variation. The training of the network while imposing cycle consistency is done by simply minimizing the cross-entropy to predict the nearest neigbhor of an input point in the embedding space.
+ Some of the empirical results are interesting, as the paper reports increased mutual information (MI) between the embeddings and the factors of variation of interest.

While I appreciate the simplicity of the approach, there are some important concerns which the paper fails to address adequately.
- The analysis of empirical results is missing, which raises many questions.
 - Experiments of Fig. 3 indicate that with one inactive factor, two of the remining five active factors are isolated, and with two inactive factors, one of the remaining four active ones is isolated. This behavior is not clear as to why it happens? Why are the other factors not isolated? What if we have other factors (the not so easy ones like scale, shape or pose) inactive, will still the background objects' factors will be isolated? This behavior should ideally be explained through further analysis experiments. Similarly for other datasets.
 - It is not clear how the temperature value (as used in Definition 2) is set. How sensitive is the performance to this parameter.
 - How is the dimensionality of the embedding space determined? How does it effect the isolation / disentangling performance? (Jha et al. 2018) seem to show that the latent space dimensionality impact disentangling performance. Could this be a reason for the other factors not being isolated in Fig. 3? The embedding dimensionality is only 2.

- Comparisons with group level supervision work
 - (Bouchacourt et al., 2018) used group level supervision for disentanglement, and would have been more appropriate for comparisons than e.g., (Jha et al. 2018). It is not clear why no comparisons were made with this work.

- There are other mistakes, possibly typographical in nature.
 - From Definition 2, it appears that the soft nearest neigbhor of l_i has larger \alpha_j for points m_j that are farther away from l_i.
 - Para before Definition 1, defines B, such that |B|=m, but it seems that it is meant to be n.

My recommendation is to reject this paper, primarily because of the lack of analysis and ablative experiments, as well as comparisons to the related approach of (Bouchacourt et al. 2018). It is unclear why certain factors get isolated while others do not, based on the active and inactive sets of factors. Well-designed analysis experiments will perhaps bring about some additional insights.

I believe the paper will make a far more compelling case if there are analysis experiments presenting the strengths of the approach that provides insights into why certain factors are easily isolated and others are not.

---

> ### Author Response · Authors · 2020-11-20
> **Expanded analysis and discussion around factor isolation**
>
> We thank the reviewer for the time taken to carefully read our paper and have followed the suggestion to make a comparison to the related method of Bouchacourt et al. (2018).
>
> *I believe the paper will make a far more compelling case if there are analysis experiments presenting the strengths of the approach that provides insights into why certain factors are easily isolated and others are not.*
>
> We have focused much of the added material to shed more light on factor isolation:
> - Revised Discussion
> - Appx. B: Shapes3D factors embedded in higher dimensions
> - Appx. C: More constrained settings for Shapes3D, getting at the ‘tougher’ geometric factors
> - Appx. D: Why a single factor of variation isn’t enough for the ACC loss
>
> *Experiments of Fig. 3... Why are the other factors not isolated?*
>
> Solving the correspondence task by way of the ACC loss does not require extracting every factor of variation.  The factors of variation differ in salience with respect to the embedding network, as shown by the mutual information results of Fig. 3 where the hue factors are consistently more easily extracted.
>
> - Added to the Discussion (P9): “While isolating multiple active factors may lower the ACC loss on average, factors differ in salience. The hue-related generative factors of Shapes3D appear easier for the specific network to identify, so once a correspondence utilizing these factors is found, training effectively ceases. Similarly, nuisance factors of variation in the images of cars and chairs are easier for a network to identify than camera pose, which is why double augmentation helped to encourage the network to isolate pose.”
> - Added numerical results to Appx. D which show the ACC loss decreases by finding more independent factors of variation but depends on the size of the training sets.
>
> *It is not clear how the temperature value (as used in Definition 2) is set. How sensitive is the performance to this parameter.*
>
> The temperature sets the scale of lengths in embedding space, meaning all interpoint distances will simply be expanded or contracted by the embedding network to match the temperature if using anything derived from Euclidean distance.  Cosine similarity is bounded, however, making temperature a meaningful hyperparameter for the pose estimation experiments.
> - Added to Appx. A, implementation specifics: “We used squared L2 distance as the embedding space metric and a temperature of 1, though as long as the length scale set by the temperature is larger than the initial point spread from the randomly initialized network, it does not seem to matter.”
>
> *How is the dimensionality of the embedding space determined? How does it effect the isolation / disentangling performance? … Could this be a reason for the other factors not being isolated in Fig. 3?*
>
> The Shapes3D experiments of Fig. 3 used two dimensions so that mutual information could be reliably measured, but we show the same behavior (though probed differently) occurs in 64-dimensional embedding space (Appx. B).  We find that higher dimensions help with training even though the learned embedding distribution is generally relatively low dimensional as measured by PCA.
> - Added reproduction of Fig. 3 experiments in higher dimensions, Appx. B
> - Added dimensionality ablative study for pose estimation, Appx. E.
>
> *Comparisons with group level supervision work, e.g. (Bouchacourt et al., 2018)*
>
> - Implemented ML-VAE (Bouchacourt 2018) referring to the code posted at https://github.com/DianeBouchacourt/multi-level-vae and added the results to Tab. 1 (pose estimation) and Appx. F (digit retrieval).
> - Added to Sec. 4 (P7): “We compare to the representations yielded by two VAE-based approaches which utilize grouped data to separate factors of variation: CC-VAE (Jha et al., 2018) in Fig. 4 and ML-VAE  (Bouchacourt et al., 2018) in Appx. F.”
> - Added to Sec. 4 (P8): “The significant difference between ACC and the generative approaches underscores the importance of meaningfully incorporating unannotated real images during training; there is no simple means to do so with either VAE-based method.”
>
> *There are other mistakes, possibly typographical in nature.*
>
> - Corrected the two that were raised and checked over the rest of the paper.

---

### Official Review · AnonReviewer2 · 2020-10-28
**A feature learning method for grouped data. The method seems not supported enough by motivation nor results.**

**Rating:** 4
**Confidence:** 3

**Review:**

The submission proposes a method for representation learning in a setting where data is annotated by set membership (i.e. grouped data). More specifically, the authors aim to extract representations of the factors of variations that are shared across groups.
To do so, the proposed method embeds different sets into a shared latent space using a learning objective called Affinity Cycle Consistency (ACC). ACC imposes a soft version of the cycle consistency on the nearest neighbourg relationship between the two learned sets of embeddings.
Using this method, the authors show in experiments on 3DShapes and MNIST that the inactive features (features that fixed in each set) are removed from the embeddings that, in turn, better encodes for the active features (features that vary inside each set).
They also show how their methods can be used to accurately estimate the pose of objects (cars and chairs) in unlabeled real pictures by aligning them with sets of synthetic images of cars grouped by pose.


################################################

Strong points:

-Representation learning for grouped data is a relevant topic, for which the submission proposes a simple and effective method.

-The authors present a practical use case in section 4.3, showing how the method can help tackle the domain gap problem. Also, experiments presented in Figure 3 uncovers some very intriguing properties that might be worth exploring further.

-The paper is well organized, formalism is clear and related work is informative.


Weaknesses:

-ACC, as noted by the authors, is very similar to the previously introduced Temporal Cycle Consistency. The main contribution of this work is to provide empirical results when applied to a more general context.

-The paper lacks crucial discussions about why ACC allows learning a good alignment between sets in the general context. The choice of ACC seems arbitrary, as I find unclear how ACC relates to latent spaces alignment.

-While the experimental results are intriguing, they aren't very convincing in terms of the usefulness of the proposed method. Results in section 4.3 are obtained using a 2-dimensional latent space, which is useful for visualization but is a very odd choice for practical uses. Section 4.5 presents an interesting application but the fact that the authors had to use a complex pipeline for these relatively simple images raises concerns about how well the method can generalize to other problems or scale to more difficult images.


################################################

Rating motivation:

The paper advocates for the use of ACC for representation learning but doesn't provide an explanation about why the method should work. This makes it difficult to assess how much of the results can be attributed to the use of ACC and how much is due to other inductive biases. Also, while the car experiments are interesting, the results by themselves aren't convincing enough.

I might change my evaluation if this point is clarified (either via discussion or by additional control experiments).


################################################

Questions and minor remarks:

-Can the authors comment on the importance of the size of the embeddings? Does the capacity bottleneck explain that geometric features are less prominent in figure 3c? What happens when using larger embedding?

-Can the author confirm that phi:A->L and phi:B->M are implemented as two different convolutional networks? The paper is a little ambiguous.

-The method is simple but reproducing would probably need some additional implementation details (architectures, optimization method, hyper-parameters...)

-While the active/inactive features formalism is clear, it still might help to illustrate with examples how some classical tasks (for instance in domain transfer) fit in.

---

> ### Author Response · Authors · 2020-11-20
> **Expanded discussion and provided additional control experiments**
>
> We thank the reviewer for their feedback following a close reading of our paper, and address raised points below.
>
> *... very similar to the previously introduced Temporal Cycle Consistency.*
>
> Whereas the aim of Temporal Cycle Consistency was to introduce a method to align video sequences of a given action, it is our express goal to show the generality of the method and bring it into the active research direction of optimally leveraging weak supervision to extract useful representations.  We develop the language and framework necessary to cast the methodology of Dwibedi 2019 into a far broader range of scenarios.
>
> In addition to the contribution of generalization, we introduce two modifications to the method which further serve its generalization and dissemination.  Unannotated data from a similar distribution can be incorporated, and the effect it has on the pose estimation task in Tab. 1 (the difference between ACC and the VAE-based approaches) is enormous.  We also introduce the double augmentation scheme to serve as another tool for operating on factors of variation in a dataset.
>
> *The paper advocates for the use of ACC for representation learning but doesn't provide an explanation about why the method should work.*
>
> We have focused much of the additional material in the revision on providing intuition about why and how ACC works.
> - Rewritten Discussion Sec.
> - Added Appx. B: repeats the Shapes3D experiments in higher dimensions
> - Added Appx. D: employs Monte Carlo numerics to show why multiple factors of variation are extracted when only one should be needed to find correspondence
> - Included GIF of the training evolution of MNIST, which helps visualize ACC (supplemental material)
>
> *While the experimental results are intriguing, they aren't very convincing in terms of the usefulness of the proposed method….  raises concerns about how well the method can generalize to other problems or scale to more difficult images.*
>
> We have included an additional pose estimation experiment (Tab. 2 in Sec. 4.3) which further showcases the ability of ACC to isolate pose and enhance performance on a challenging real-world task.  We add a spherical regression head (using the framework of Liao et al. 2019) on top of the ACC embedding space and operate in the data setting where pose annotations are available for synthetic images but not real, with the test set from Pascal3D+.
>
> The conclusion is that ACC allows the incorporation of unannotated real images which significantly improves the accuracy compared to the baseline of regression trained solely on synthetic images.
> - The median angular error drops from 12.3 to 9.3 deg  for Pascal3d+ cars, and from 30.8 down to 26.0 deg for chairs. (Fig. 6, Tab. 2).
>
> *Results in Sec. 4.3 are obtained using a 2-dimensional latent space, which is useful for visualization but is a very odd choice for practical uses.*
>
> The 2-dimensional latent space was chosen to facilitate the mutual information measurement.  We reproduce the results in higher dimensional embedding spaces in Appx. B.
> - Added Appx. B: repeats the Shapes3D experiments in higher dimensions
> - Added to the Discussion (P9): “[The factor isolation behavior of Fig 3] can be partly attributed to the low dimensionality of the embeddings -- a design choice to allow the measurement of mutual information, which is notoriously problematic in higher dimensions -- though we show in Appx. B that the effect is also present for 64-dimensional embeddings.”
>
> *the authors had to use a complex pipeline for these relatively simple images…*
>
> We assume that the complex pipeline referred to was, at least in part, the nearest neighbor lookup used for the results of Tab. 1 in order to extract pose information.  This was by design: this scenario was meant to showcase the performance of ACC trained without a single ground truth annotation for the quantity of interest.
>
> *Can the authors comment on the importance of the size of the embeddings? … What happens when using larger embedding?*
>
> We used two dimensions for the Shapes3D experiments in order to facilitate the mutual information measurements. In experiments added to Appx. B, we probe 4, 16, and 64 dimensional embeddings by using a simple FF network to classify each of the six generative factors given the embedding as input. In higher dimensions, factor of variation isolation is subtly different, and connected to length scales over which the information is encoded.
> - Added Appx. B: reproduction of Fig 3 experiments in with higher-dimensional embedding space
> - Added Appx. E: ablation studies for the pose regression.
>
> *Can the author confirm that phi:A->L and phi:B->M are implemented as two different convolutional networks?*
>
> No, there is only one network (shared weights) which does the embedding for both input sets.
> - Added to Methods (P4): “Functionally, we parameterize \phi with the same neural network for both [input sets] A and B.”

---

> > ### Author Response · Authors · 2020-11-20
> > **Expanded discussion and provided additional control experiments (continued)**
> >
> > *The method is simple but reproducing would probably need some additional implementation details (architectures, optimization method, hyper-parameters...)*
> > - Added implementation specifics to Appx. A
> > - Including iPython notebook with code to run the MNIST digit style isolation
> > - We plan to post the code for the Shapes3D and pose estimation experiments after cleaning it up.
> >
> >
> > *While the active/inactive features formalism is clear, it still might help to illustrate with examples how some classical tasks (for instance in domain transfer) fit in.*
> >
> > We have tried to elucidate the distinction between active and inactive factors in more settings.
> > - Added to Discussion (P9): “A correspondence can be made when each element is embedded according to only a single active factor of variation common to both sets. This was the case for Dwibedi (2019), where the progression of an action (e.g., bowling) was the only active factor of variation (with scene specifics being inactive, fixed per video).”

---

### Author Response · Authors · 2020-11-12
**Thank you reviewers**

We thank the reviewers for the time spent carefully reading our manuscript and for the suggestions which will greatly help us improve our paper.  We are in the process of revising and will post an updated version of the paper in about a week, with responses to each review shortly thereafter.

---

### Author Response · Authors · 2020-11-20
**More insight, more experimental backing**

We thank the reviewers for the detailed and constructive feedback. We appreciate the positive feedback that this is a “difficult and important problem” (R1) to which we propose a “simple and effective method” (R2, R3); our results uncover “intriguing properties” worthy of further exploration (R2); and that the manuscript contains clear formalism (R2), is “well written” (R4), and has “informative related work” (R2). We address some of the common concerns below:

**Insufficient analysis of the factors of variation that are extracted by ACC (R1, R2, R3):**

Affinity cycle consistency isolates active factors of variation in learned representations, but generally not with respect to all of them.  We devoted the majority of the original submission to demonstrating that ACC works in a wide variety of scenarios and developing a language for describing the performance.  The next natural question which arises, after the framework has been established and the method is shown to work, is why: specifically, why some factors but not others?  The fact that so many of the reviewers had this question is evidence that our case was successfully made.

Most of the added material in the resubmission focuses on this topic, to leave the reader with more insight about how the factor of variation isolation works in practice.  Please see the change list below with specifics.

**Insufficient technical novelty (R4, R2):**

To clarify, the primary contribution of our paper is to elevate a narrow-purpose method for temporally aligning video sequences to a general framework for operating on factors of variation given generic data groupings.  Beyond generalizing affinity cycle consistency and opening the door for myriad new use cases, we also introduce two impactful innovations: the incorporation of unannotated and possibly out of domain data, and greater control over factors of variation with the double augmentation modification to the loss.  We think the sum total contribution of this work to the field of representation learning will be significant.

**More convincing pose estimation results (R2):**

The main takeaway from the pose estimation experiments was how impactful it is that ACC can factor in unannotated real images and meaningfully connect them to the synthetic images for which there is set supervision.  There is no analogous means of doing so for the comparable VAE-based approach, and the performance gap is dramatic.  We have clarified this point in the text, and strengthened the claim by comparing to a second relevant VAE-approach (Bouchacourt 2018) as suggested by R3 (Tab. 1).

We also add to the manuscript pose estimation results in a more realistic setting, of pose regression supervised on synthetic pose annotations and where the ACC loss is optimized on an intermediate embedding space.  The unique ability of ACC to incorporate unannotated data from a different domain again leads to a significant boost in accuracy (Tab. 2).

**Did the 2D embedding space of the Shapes3D experiments affect performance? (R2, R3):**

We reproduced the Shapes3D experiments in higher dimensional embedding spaces and found the same qualitative behavior, though with some subtle differences (Appx. B).

**Reproducibility (R1, R2):**

We add implementation specifics (Appx. A) and include in our resubmission code to reproduce the MNIST digit style isolation results.  We also plan to release the code for the Shapes3D and pose estimation experiments.

**List of changes:**

- Rewritten Discussion
- Appx. A: Implementation details
- Appx. B: Shapes3D experiments in higher dimensional embedding space
- Appx. C: More inactive factors in the Shapes3D experiments
- Appx. D: Motivating why multiple independent factors of variation are isolated in each of the experiments, with Monte Carlo experiment
- Appx. E: Ablative studies for pose estimation problem
- Appx. F: Extended comparison between ACC and generative methods for MNIST experiment
- Tab. 2 with supervised pose regression improved by ACC and the incorporation of unannotated real images
- Fig. 6 with schematic for the pose regression experiment
- iPython notebook to reproduce the MNIST digit style isolation results and a GIF showing the evolution of embeddings during training (supplemental)

---

### Decision · Program_Chairs · 2021-01-07
**Final Decision**

**Decision:**

Reject

**Comment:**

This paper proposes to employ affinity cycle consistency(ACC) for extracting active (or shared) factors of variation across groups. Experiments shows how ACC works in various scenarios.

Pros:
- The problem is important and relevant.
- The paper is well written.
- The proposed method is simple and effective.

Cons:
- The experimental section is weak:
 It lacks an ablation to validate the contribution of ACC and discussion on
  why the method works and the scalability of the proposed method to more complex cases.
- The novelty is limited because the proposed ACC is similar to previous work temporal cycle consistency(TCC).
- The paper missed some implementation details and could be difficult to reproduce without code
 provided.

Reviewers raised the concerns listed in Cons. The authors conducted additional experiments and added more discussions on the experimental results in the revised paper. The authors also explained that ACC is more general than TCC. However, the reviewers were not convinced by the rebuttal and kept their original ratings.

Due to the two main weaknesses -- limited novelty and weak experimental analysis, I recommend reject.